# Sex-Moderated Divergence between Adult Child and Parental Dietary Behavior Patterns in Relation to Body Mass Condition—Evaluating the Mediating Role of Physical Activity: A Cross-Sectional Study

**DOI:** 10.3390/nu16142256

**Published:** 2024-07-13

**Authors:** Jarosław Domaradzki, Małgorzata Renata Słowińska-Lisowska

**Affiliations:** 1Department of Biostructure, Wroclaw University of Health and Sport Sciences, 51-612 Wroclaw, Poland; 2Department of Biological and Medical Foundations of Sport, Wroclaw University of Health and Sport Sciences, 51-612 Wroclaw, Poland; malgorzata.slowinska-lisowska@awf.wroc.pl

**Keywords:** dietary patterns, parent–child patterns, dietary intake, physical activity, adolescence, overweight

## Abstract

The main objective of this study was to explore the dietary behaviors of parents and their adult children, focusing on patterns, potential intrinsic and extrinsic predictors of body mass, and determinants of becoming overweight. Non-probability, cross-sectional sampling was used to select participants from a university student population. Young adults (19–21 years of age, n = 144) and their parents were examined. The data of those family pairs with complete sets of results were used. Dietary patterns and physical activity were assessed with questionnaires (QEB and IPAQ), and body height, weight measurements, and body mass indexes were calculated. A cophylogenetic approach with tanglegrams and heatmaps was used to study patterns, while predictors of body mass index were identified using multiple linear regression, stepwise logistic regression, and mediation analysis procedures. Cophenetic statistics confirmed significant incongruence between fathers and sons, confirmed by Baker’s Gamma correlation (r_BG_ = 0.23, *p* = 0.021), and mothers and daughters (r_BG_ = 0.26, *p* = 0.030). The relationships between the dietary patterns of the fathers and daughters, as well as mothers and sons, were of medium strength (r_BG_ = 0.33, *p* = 0.032, r_BG_ = 0.43, *p* = 0.031; respectively). Most of the patterns were mixed. Fast food, fried meals, alcoholic drinks, energy drinks, and sweetened beverages were associated significantly with being overweight. Significant intrinsic predictors of excessive weight in young adults were sex (b = 2.31, *p* < 0.001), PA (b = −0.02, *p* < 0.001), and eating fermented milk and curd cheese (b = −0.55, *p* = 0.024), while extrinsic (parental) predictors included eating fast food and fried meals (b = −0.44, *p* = 0.049). Both physical activity and dietary behaviors independently determined the sons’ overweight status (b = −1.25, *p* = 0.008; b = −0.04, *p* < 0.001; respectively); while only PA did in daughters (b = −0.04, *p* < 0.001). No mediating effects of physical activity were observed. Adult children and parental dietary patterns were divergent, reflecting the influence of multiple factors on a child’s dietary habits. However, this divergence is moderated by sex. Reciprocal interactions between dietary intake—particularly positive dyads such as fruits and vegetables, fermented milk, and curd cheese—and physical activity significantly impacted children’s body mass index (BMI). The study of dietary patterns in conjunction with physical activity (both as independent determinants), particularly in relation to the link between overweight/obese children and overweight/obese parents, presents a separate challenge.

## 1. Introduction

There is mounting evidence indicating that the number of overweight and obese boys and girls aged 5–19 years old has increased by approximately tenfold from 1975 to 2016 [1]. Adolescence is a critical period of growth and development, marked by significant physical, psychological, and behavioral changes. Weight monitoring at this stage is essential due to its impact on one’s overall health and well-being [2]. Understanding patterns of weight development during adolescence and identifying risk factors associated with becoming overweight are critical to designing effective interventions [3].

Good dietary habits need to be developed during childhood to improve short-term health, but also, over the long period of life, to avoid carrying unhealthy habits into adulthood [4]. Bad dietary habits, acquired in childhood or adolescence, are strongly associated with negative health outcomes throughout a lifetime, such as an increased risk of cardiovascular disease [5,6]. The key predictor of a child’s dietary intake is the effect of parental diet [7,8]. The main areas of this research studied the relationship between positive dietary behaviors, e.g., fruit, vegetable, and legume consumption [9], and different food groups [10,11,12,13]. Several studies have demonstrated a positive association between dietary diversity and nutrient adequacy [14,15,16,17]. Individuals who consume a diverse range of foods are more likely to obtain a broader spectrum of essential nutrients, which can contribute to better overall nutritional status and health, and is related to children and parents’ dietary similarities [18,19]. The eating behavior of parents can influence the behavior of their children in adulthood. Moreover, Semmler et al., 2009 [20] and Keane et al., 2012 [21] suggested that childhood obesity reflects the parent–child similarities in eating habits, and is associated with parental weight status. However, there is still no consensus on which role models are stronger, whether mother–child or father–child [22].

The risk of excessive weight and the prevalence of being overweight or obese is greater because of the increasing lack of physical activity (PA). PA is defined as any bodily movement produced by the skeletal muscles that results in energy expenditure [23]. The health benefits of PA are well known and no longer need to be proved. The relationship between PA and dietary behaviors has been studied many times [24,25,26]. However, it is still unknown if the relationship between PA and dietary behaviors has a synergistic or additive effect on body weight.

Research into the relationship between lifestyles, particularly dietary patterns, and obesity risk is particularly important now, in post-pandemic times intense increases in overweight and obesity rates have been observed. This global trend is also present in Poland. Large Polish nationwide surveys (n = 14.044 students, 13–19 years old) have clearly indicated that significantly increasing percentages of Polish adolescents exceed normal body weight [27,28]. Excessive weight was observed in 18.2% of adolescents, where being overweight was found in 11.6% and obesity in 6.6% of participants. The phenomenon becomes more important in the context of pandemic-induced isolation and lifestyle changes. Deterioration of dietary habits in relation to changes in PA has also been observed previously in Polish students [29].

There are only a few studies related to the complex study of the dietary and PA patterns in late adolescents and young adults (18+ years of age) [29,30] in the Polish population, specifically in the population of students who recently started schooling. Moreover, there is a lack of studies presenting the relationships and correlations between dietary behaviors in adult children and their parents. A valuable solution is using tanglegrams between the two types of patterns. This cophylogenetic (with cophenetic statistics) approach is useful, albeit rarely used in the study of dietary patterns [29]. What is more, to date, there are only a few studies evaluating the associations between the prevalence of overweight parents and their adult children. Additionally, there are no studies examining the role of PA, which may be directly related to excessive weight or may potentially mediate the relationship between dietary behaviors and weight status.

Therefore, the main objective of this work was to study parent–adult child dietary behaviors regarding patterns and potential intrinsic and extrinsic predictors of becoming overweight. Specifically, the work aimed to (1) explore the similarities and dissimilarities in pairwise parent–child dietary patterns, (2) determine intrinsic (children) and extrinsic (parental) predictors of a young adult’s excessive body mass by assessing the potential mediating role of PA and inactivity in these associations.

## 2. Materials and Methods

### 2.1. Study Design

Non-probability, cross-sectional sampling was used to select participants from a university student population. Examinations were conducted in 2023 at the Wroclaw University of Health and Sport Sciences. Full details of the study design have been published previously [29]; however, for another group of participants. Even though the procedures were the same, students were asked to complete an online survey and invite their parents to cooperate and complete the same dietary and PA questionnaires. In this study, data from parent–child pairs were used. In addition, parents self-reported their body height and weight, while children were precisely measured by one of the authors.

### 2.2. Ethics

The study was conducted according to the guidelines of the Declaration of Helsinki [31] and approved by the Senate Research Ethical Board of Wroclaw University of Health and Sport Sciences (consent number 13/2022, date of approval: 28 March 2022).

All participants were asked to provide informed consent (in an online form) prior to the study, and the purpose and characteristics of the research were explained.

### 2.3. Sample Size

The main statistical approach conducted in this article was cophylogenetic tanglegrams and cluster analyses. In the case of the cluster analysis, Dalmaijer et al. [32] recommend, as a rule of thumb, to have a minimum of 30 participants per anticipated subgroup. Four clusters were assumed in each pairwise comparison (parent–child with sex variation). Therefore, a minimum of 120 males and 120 females needed to be recruited. In this study, 144 adult children, together with 144 parents, were evaluated.

### 2.4. Participants

Participants included 288 healthy individuals who were in family pairs: 46 were mothers and daughters, 36 were mothers and sons, 33 were fathers and daughters, and 29 were fathers and sons. Adult children were students in their 1st year of study at the Faculty of Physical Education and Sport and Physiotherapy in 2023 at Wroclaw University of Health and Sport Sciences. The proportions of males and females represented the true populations in these fields of study. Details of the recruiting process are presented in a flowchart (Figure 1).

Preliminary inclusion criteria were voluntary consent, stationary studies, and less than 22 years of age. The exclusion criteria were participants secondly completing their 1st year of study, sick leave (longer than 3 weeks), sports activity, and illness or injury. A total of 237 students were accepted to take part in the examinations. In this work, the main inclusion criterion was the involvement of one of the parents. Finally, data from 144 students linked with their parents were included in the analyses.

### 2.5. Data Collection

Data were collected using Family Lifestyle Patterns (FAST-PAT23), a local project that evaluated, among other things, PA, dietary behaviors, attitudes toward healthy lifestyles, and socio-economic determinants of the family lifestyle. Closed-question questionnaires were made using online Google forms immediately after an academic lecture (Human Anatomy taught by the author). Recruitment, data collection, and entry were conducted by one of the authors of this article. After completing the questionnaires, the students were successively examined (anthropometric measurements and body composition) over four consecutive weeks in March of 2023 according to a set schedule. The parent’s height and weight were self-reported in the questionnaire.

### 2.6. Questionnaire Measurements

#### 2.6.1. Dietary Characteristics

Dietary behaviors were gathered one year prior to examinations using the food frequency method, from a self-administrated Questionnaire of Eating and Behaviors (QEB) [33]. The internal reliability of the QEB was assessed as great with a Fleiss’ kappa from 0.64 to 0.84 [34]. The minimal set of 16 questions recommended in the instructions for the QEB were used in this study. The questions were separated into two groups: positive dietary behaviors and negative dietary behaviors. The frequency of consumption of each food group was expressed using 6 measurement categories: never, 1–3 times per month, once per week, several times per week, daily, and several times per day. Each category was converted to coefficients, and the frequency of consumption was expressed as times/day (never = 0, 1–3 times per month = 0.06, once per week = 0.14, several times per week = 0.5, daily = 1, and several times per day = 2). Finally, two indexes were calculated: the Pro-Healthy Diet Index (positive HDI) and the Non-Healthy Diet Index (negative HDI). Details of the questionnaire with a full list of the items have been published previously [29].

This questionnaire is one of the questionnaires indicated for studying the eating habits of adults [30]. Information about food groups, needed to calculate the pro-healthy and non-healthy indexes, is focused on food groups with a potentially beneficial or a potentially negative influence on health.

#### 2.6.2. Physical Activity

As supplementary data, PA information was gathered. The Polish version of the International Physical Activity Questionnaire (IPAQ), long-form, was used [35]. The IPAQ is a validated, well-known, and easy-to-use questionnaire. The questionnaire consisted of 11 items assessing PA that were separated into four domains measuring school or work, transportation, housework/gardening, and leisure time activity levels. The twelfth item included time spent sitting. The final results were scores from the raw data converted with specific formulas included in a guide. Details of the questionnaire with a full list of the items have been published previously [29].

### 2.7. Anthropometric and Body Composition Measurements

Anthropometric and body composition measurements were conducted in the Biokinetics Research Laboratory (part of the Central Research Laboratory) of Wroclaw University of Health and Sport Sciences. This facility has Quality Management System Certificates of PN-EN ISO 9001:2015 (certificate reg. no.: PW-15105-22X, validity: 27 May 2025) [36].

Two body height measurements were taken with an accuracy of 0.1 cm using an anthropometer (GPM Anthropological Instruments). Body weight was measured with a body composition analyzer using the InBody230 electronic tool (InBody Co., Ltd., Cerritos, CA, USA). Using body height and weight, the BMI was calculated.

### 2.8. Handling and Imputation of Missing Data

A detailed description of dealing with missing data has been presented elsewhere [29]. Briefly, there was missing data on the DI Questionnaire (n = 9) and IPAQ (n = 16). Since cluster analysis requires there to be no missing data, all measurements were preprocessed by applying multiple imputations. The algorithm used for imputations was related to the type of missing data pattern, called *missing completely at random* (MCAR). Imputation was conducted in R language using RStudio software v. 2022.7.1.554 (RStudio Team (2022), RStudio: Integrated Development Environment for R. RStudio, PBC, Boston, MA, url: http://www.rstudio.com/, accessed on 25 June 2024) with the package *mice* (v.3.14.0).

### 2.9. Statistics

At first, the shape of the data distribution was assessed. The variables identified as non-normally distributed (QEB and IPAQ results) were transformed to receive a normal shape of the distribution. Transforming data to bring them closer to normality was needed to meet the assumptions for the other statistical methods. To transform data with 0.0 values into a more normal distribution, the Yeo–Johnson power transformation was used [37]. Continuous variables were presented as means, standard deviations, and 95% confidence intervals (CIs) for the mean. Categorical variables were presented as numbers and percentages.

To compare outcomes between the groups, a one-way analysis of variance (ANOVA) was conducted with Bonferroni post hoc tests for detailed comparisons. Prior to analysis, Levene’s test was performed to assess the homogeneity of variances between the groups.

Similarities in patterns of dietary behaviors between parents and adult children were studied using tanglegrams and cophenetic statistics (derived from the cophylogenetic approach). Tanglegrams were drawn using the web tool CLINE [38], available at https://mizuguchilab.org/cline/ (accessed on 25 April 2024). Cophenetic statistics were calculated. The distances between the variables were characterized as mean values and standard deviations. Baker’s Gamma correlation coefficients (r_BG_), which are a measure of associations (similarities) between two trees of hierarchical clustering (dendrograms), were calculated with significance (*p*-value) derived from Mantel’s test. In these clustering analyses, preprocessing was based on (1) scaling the variables with normalization in the range [0, 1], (2) calculation of the Mahalanobis distances, and (3) agglomeration of the variables using Ward’s method as a linkage method. An illustration using a heatmap showed similarities between groups and dietary behaviors at the same time.

The prevalence of overweight and normal-weight participants in relation to various demographic factors (e.g., sex, family affinity, pairwise family links, etc.) was tested with χ^2^ tests. Frequencies were illustrated with balloon charts. The strength of the relationship between categories in contingency tables was assessed using the contingency coefficient (C). Moreover, similarities between groups separated based on BMI status (overweight and normal weight) in relation to positive and negative dietary behaviors were studied with factor analysis of mixed data (FAMD). FAMD is a principal component method dedicated to analyzing a data set containing both quantitative and qualitative variables [39]. Positive and negative dietary behaviors were assessed based on indexes calculated from the QEB (positive HDI and negative HDI), while more detailed analysis was conducted on the most common clusters, for all family pairs, revealed in the tanglegrams. To facilitate the selection of common items, specific functions were included in the dendextend package (for RStudio software version 2024.04.2+764): *dend_diff* and *all.equal* were used. The first function plots two trees side by side, highlighting edges unique to each tree in red (on the opposite side, the common edges are black), visually helping to make decisions. Next, the second function makes a global comparison of two or more dendrogram trees, printing the names of unique edges (based on the elimination of unique edges, the most common edges can be identified). Repeated items in these two functions were found to be the most common for all family pairs. If identified, items were included in clusters (diads, triads, or more) and were agglomerated using principal component analysis (PCA). The first factor was included in a FAMD analysis as a variable representing each cluster of positive or negative behaviors. Eigenvalues were presented in the text. Family affinity (father, son, mother, daughter) in BMI categories (overweight and normal weight) were one-hot encoded to transform categorical data into numerical data by presenting one column for each modality of each categorical variable, with 1 being if the variable takes the modality and 0 in other cases. Intrinsic (children) and extrinsic (parental) determinants of body weight were analyzed with stepwise backward multiple linear regression. The dependent variable included the adult children’s BMI (as a continuous variable), while the independent variable included the (intrinsic) student’s sex (male was coded as 1), positive and negative dietary behaviors (revealed in dietary patterns), PA (IPAQ scores), and inactivity (sitting time [min/week]). In the second model, the variables included (extrinsic) parental sex (male was coded as 1), positive and negative dietary behaviors, PA, inactivity (sitting time), and BMI.

The study of overweight determinants was conducted using multiple logistic regression. Separate models were built for males and females. The goodness of the model’s fit was assessed using the chi-square of goodness of fit. Regression coefficients and odds ratios (ORs) were calculated. The dependent variable included the adult children’s BMI (overweight was coded as 1, normal weight as 0), while the independent variables included dietary intake, PA, and sitting time. In this analysis, dietary behaviors (negative and positive components) were combined with PCA into one index using the first factor. In the case of significant dietary behaviors and PA and/or sitting time (inactivity), the mediation role of the time spent active or inactive was examined. The original Baron and Kenny (1986) [40] proposition was used, with MacKinnon and Cox (2012) [41] modifications for the categorical variables. Logistic regression was employed with procedures described by Newsom [42]. This analysis answered the question about the possible importance and hierarchy of dietary behaviors and PA in shaping the overweight population.

The significance level for all statistical tests and procedures was set at an α-value equal to 0.05. Calculations were conducted using RStudio with the following additional R packages: *dendextend* [43], phytools [44], plyr [45], ape [46], viridis [47], BiocManager [48], and ade4 [49], Statistica 13.5 (StatSoft Poland 2023, Cracow, Poland), and the CLINE web-tool [38].

## 3. Results

### 3.1. Sample Characteristics

Table 1 and Table 2 display the characteristics of the study sample (adult children and their parents, respectively). Only total values of the positive and negative dietary behaviors, overall IPAQ, and sitting time results are shown.

Within the group of adult children (Table 1), the general affinity factor significantly affected body height (F = 76.40, *p* < 0.001), weight (F = 44.96, *p* < 0.001), BMI (F = 44.96, *p* < 0.001), positive and negative dietary behaviors (F = 24.46, *p* < 0.001; F = 36.15, *p* < 0.001, respectively), and PA (F = 36.15, *p* < 0.001). Overall, boys were significantly taller and heavier than girls (*p* < 0.001). However, only sons of mothers differed in BMI from daughters of mothers (*p* = 0.004) and fathers (*p* = 0.009). Concerning positive dietary behaviors, all pairwise comparisons were significant (*p* < 0.05), except for the difference between sons of fathers and sons of mothers. Regarding negative dietary behaviors, sex differences were significant (*p* < 0.001), while there were no differences between sons (fathers and mothers) and daughters (fathers and mothers). Sons of mothers had the highest PA levels (transformed into IPAQ values) compared to the rest of the groups, and the differences in comparison to the daughters of mothers and daughters of fathers were significant (*p* < 0.001 and *p* = 0.034, respectively). However, there were no differences in sitting time.

Within the group of parents (Table 2), the general affinity factor significantly affected body height (F = 37.50, *p* < 0.001), weight (F = 13.29, *p* < 0.001), positive and negative dietary behaviors (F = 39.45, *p* < 0.001; F = 29.29, *p* < 0.001, respectively). Overall, fathers (of the sons and daughters) compared to mothers were significantly taller (*p* < 0.001) and heavier (*p* < 0.001). However, there were no significant differences in BMI. Concerning positive and negative dietary behaviors, fathers did not differ from each other, mothers did not differ from each other, and differences applied to all comparisons between fathers and mothers were significant (*p* < 0.001). There were no differences in the IPAQ and sitting time results.

### 3.2. Patterns of Dietary Behaviors in Family Affinity Comparisons

The two tanglegrams of the dendrograms generated concerning the dietary behaviors between family pairs (fathers and sons/daughters, mothers and sons/daughters) are shown in Figure 1. The fathers of sons’ dietary behaviors (Figure 2A) are grouped into three huge clusters and were individually connected by the number of meals during the day. Specifically, there were two negative dyads: 1st—alcoholic and energy drink consumption and 2nd—sweetened beverages and canned meals. Sons did not imitate the behaviors of their fathers, and there was no similar pattern between fathers and sons. The son’s behaviors are mixed in nature. Specifically, there were two dyads: positive—consisting of vegetables and fruits and negative—consisting of energy drinks and fast food meals. A weak association in the father–son structure of dietary behaviors was confirmed by cophenetic statistics. The average father and son distances were 5.51 (±5.1) and 5.68 (±5.1). The Baker’s Gamma correlation coefficient for the two trees (also known as the Goodman–Kruskal gamma index) showed that the tree’s topology was incongruent; however, the small relationship was significant (r_BG_ = 0.23, *p* = 0.021).

Comparing the tanglegrams of fathers and daughters (Figure 2B), more similar patterns in dietary behaviors were observed. However, the nature of behaviors is mixed and not identical between fathers and daughters. The father’s behaviors are mixed and contain a negative dyad consisting of fast food and fried meals, a positive dyad consisting of vegetables and fruits, and a mixed triad consisting of legumes, fish, and canned meals. In comparison, daughters have the same mixed triad and positive dyads. Daughters imitate their father’s positive behaviors to a limited extent. The average father and daughter distances were longer compared to the father–son picture (6.05 ± 5.3 and 6.05 ± 5.4, respectively). The strength of associations between the patterns was medium and statistically significant (r_BG_ = 0.33, *p* = 0.032).

Comparing tanglegrams for mothers and sons (Figure 3A), more similar patterns in dietary behaviors were observed. However, it was limited to one cluster. In both, the mother’s and son’s dietary behaviors were composed of five positive items: one dyad—vegetables and fruits and one triad—fermented milk, milk, and wholegrain bread. Sons imitate their mother’s positive behaviors to a limited extent. The average mother and son distances were longer compared to the father–son picture (6.33 ± 5.6 and 6.45 ± 5.5). Baker’s Gamma correlation coefficient shows slightly greater congruence, which was also significant (r_BG_ = 0.43, *p* = 0.031).

The last comparison is related to the mother’s–daughter’s dietary behavior patterns (Figure 3B). Both mother and daughter behaviors are mixed, combining positive and negative behaviors. Specific relations are related to three dyads: first negative—consisting of alcoholic drinks and sweetened beverages, second—consisting of energy drinks and fast food, and third—positive, consisting of vegetables and fruits. Daughters imitated mothers to a limited extent. The average mother and daughter distances were the longest among all presented comparisons (7.23 ± 6.3 and 7.69 ± 5.7). Similarity between the two trees of hierarchical clustering was low, confirming a small, but significant, value of the Baker’s Gamma correlation coefficient (r_BG_ = 0.26, *p* = 0.030).

A summary of the detailed comparisons is presented in Figure 4, which shows the relationships between all eight groups and dietary behaviors. Two-way clustering analysis examined the relationship between the dietary patterns with the structure of the family group connections. First of all, the divergence in family patterns was revealed, confirmed by the dissimilarity between parent–child patterns. However, close connections between the fathers and daughters, as well as mothers and sons, allow us to conclude that the sex moderation role had dissimilarities. On the other hand, strong associations between groups (independent of family connections) and some dietary behaviors (particularly negative behaviors like consumption of fast food and fried meals, energy and alcoholic drinks, and sweetened beverages) lead to questions about potential excess weight related to such dietary behaviors or whether the positive behaviors (consumption of fermented milk, vegetables and fruits, and curd cheese) are predictors of normal weight.

### 3.3. Prevalence of Overweight among Parents and Adult Children

Out of the 144 adult children, 34 (23.61%) were overweight, while out of the 144 parents, 74 (51.39%) were overweight. The difference in frequencies was statistically significant; however, the strength of associations between the categories was small (χ^2^ = 23.70, *p* < 0.001, OR = 0.29, 95% CI range: 0.18–0.48, C = 0.28). Studying the prevalence of overweight and normal weight participants, regarding sex, the frequencies were as follows. Out of the 70 daughters, 10 (12.66%) were overweight, while out of the 82 mothers, 37 (45.12%) were overweight. The difference in proportions between both groups of women was statistically significant, and the relationship was medium (χ^2^ = 20.51, *p* < 0.001, OR = 0.18, 95% CI range: 0.08–0.39, C = 0.34). Meanwhile, out of the 65 sons, 24 (36.92%) were overweight, while out of the 62 fathers, 37 (59.68%) were also overweight. This difference in proportions was statistically significant, and the strength was small (χ^2^ = 6.58, *p* = 0.010, OR = 0.40, 95% CI range: 0.19–0.81, C = 0.22). Generally, higher frequencies of overweight were seen among male participants compared to female participants, as well as in parents compared to adult children, and were statistically significant and of medium strength (χ^2^ = 35.85, *p* < 0.001, OR— not calculated, C = 0.35).

### 3.4. Conformity of Overweight in Families—Parent–Child Pairwise Matching

The conformity of overweight and normal-weight participants in pairwise parent–adult child associations is presented in Figure 5. It can be seen that in all pairwise connections, more pairs were non-matched. The most unmatched pairs were observed between fathers and daughters (four times more unmatched in comparison to matched). The lowest frequency of matched pairs had one of the highest contributions to the chi-squared test scores (26.6%). The second-most common unmatched pairs were related to mothers and daughters (three times more unmatched in comparison to matched). Despite differences in proportions between family pairs, the results of the chi-square test were not significant (χ^2^ = 4.62, *p* = 202, C = 0.18). However, general differences in proportions between all matched and unmatched family pairs, using the chi-square goodness of fit test, were significant (χ^2^ = 26.69, *p* < 0.001). It confirmed the lack of BMI status patterns in families and led to questions about the relationship between BMI status and specific positive and negative dietary behaviors in general, independent of family dependences.

### 3.5. Specific Dietary Behaviors in Relation to Being Overweight

The next step of the analysis was to assess the relationships between family pairs, overweight, and specific dietary behaviors. The analysis was conducted with specific dietary behaviors, observed in Section 3.2, as clusters most common in all family pairs. The strategy of analysis was based on the FAMD. The results are presented in Figure 6.

The starting point was to agglomerate information containing the most common dietary behaviors of family pairs. The identified positive and negative items very often coexisted as clusters of dyads or triads. The PCA method was used to combine each cluster into common synthetic indexes. The consumption of vegetables and fruits was joined together (the *vege.fruits* variable, positive dyad, eigenvalue of the first factor: 1.67, variance explained by the first factor: 83.60%), fermented milk and curd cheese were joined together (the *ferm.curd* variable, positive dyad, eigenvalue of the first factor: 1.14, variance explained by the first factor: 57.04%), fast food and fried meals were joined together (the *fast.fried* variable, negative dyad, eigenvalue of the first factor: 1.28, variance explained by the first factor: 63.90%), and sweetened beverages, alcoholic drinks, and energy drinks were joined together (the *swbev.alco.ener* variable, negative triad, eigenvalue of the first factor: 1.67, variance explained by the first factor: 55.77%).

Next, FAMD was conducted with categorical variables: family relationships (father, son, mother, daughter) and overweight status (yes/no) and continuous variables: *vege.fruits*, *ferm.curd*, *fast.fried*, *swbev.alco.ener*. The eigenvalue for the first factor was 2.84 with a variance of 31.54%, while the eigenvalue for the second factor was 1.58 with a variance of 17.51%. The first dimension was defined with negative dietary behaviors like frequently drinking sweetened beverages, energy drinks, and alcoholic drinks (24.36% of the contribution to this dimension) and eating fast food and fried meals (11.00% of the contribution). These behaviors were specific, particularly for fathers (19.30% of the contribution) and somewhat for sons (9.64% of the contribution) and can lead to being overweight (18.53% of the contribution). However, fathers and sons, negatively self-feeding, consumed a relatively lot of vegetables and fruits (21.42% of the contribution to the first factor). The third dimension more specifically defined mothers (20.31% of the contribution to the third dimension), daughters (34.38% of the contribution) who frequently eat fermented milk and curd cheese (52.41% of the contribution) and vegetables and fruits (16.19% of the contribution).

Visually, similarities are presented in Figure 6, which shows cluster analysis derived from FAMD. Two clusters are clearly visible. The first consists of two clads: fathers who drink a lot of sweetened beverages, energy drinks, and alcoholic drinks, who are mostly overweight. This clad is closely connected with the second clad: sons who eat fast food and fried meals. The second cluster contains mothers and daughters, who frequently eat fermented milk and curd cheese.

This analysis confirmed overweight to be more prevalent among fathers and sons; however, it indicated a relationship with negative dietary habits.

### 3.6. Intrinsic and Extrinsic Predictors of Body Weight of Adult Children’s BMI

The next step in the analysis was the assessment of the potential intrinsic and extrinsic determinants of the adult child’s BMI. Two multiple stepwise backward models were constructed. The results are presented in Table 3 (*b* coefficients with 95% C, *p*-values, and accuracy statistics: R^2^, RMSE, MAE, and AIC).

First, seven independent variables, related to sons and daughters themselves, were tested: sex, consumption of vegetables and fruits (*vege.fruits*), fermented milk, curd cheese (*ferm.curd*), fast food, fried meals (*fast.fried*), and sweetened beverages, alcoholic drinks, and energy drinks (*swbev.alco.ener*), as well as PA (IPAQ scores) and inactivity (sitting time). The most significant (or close to significance, *p* < 0.1) variables were identified. In the case of intrinsic determinants, the determining factors of a child’s BMI were sex, *ferm.curd*, and IPAQ. Males did have higher BMIs; however, body weight compositions were not taken into account. Eating fermented milk and curd cheese is related to a lower BMI (negative coefficient near *ferm.curd*). The lower the PA (negative IPAQ coefficient), the higher the BMI. In addition, inactivity (sitting time) trended towards significance, where the more time spent sitting (positive sitting coefficient), the higher the BMI. The model had medium accuracy and was confirmed with adequate statistics (R^2^ = 0.18, RMSE = 2.85, and MAE = 2.25, Table 3).

On the other hand, the model for extrinsic (parental) behaviors showed a significant relationship between the adult children’s BMI and consumption of fast food and fried meals by the parents. The more the parents consumed, the higher the children’s BMI. Very close to significance was also the consumption of vegetables and fruits and parental PA (*p* = 0.111, *p* = 0.112, respectively, Table 3). The accuracy of the model was worse fitted than for intrinsic determinants (R^2^ = 0.10, RMSE = 2.90, and MAE = 2.26) and generally was worse compared to the previous model, confirmed with lower AIC statistics for the intrinsic model (AIC = 279.84, AIC = 311.97).

### 3.7. The Role of Physical Activity and Inactivity in Relation to Dietary Behaviors and Weight Conditions in Adult Children

Previous analysis showed significant effects of the children’s dietary behaviors and PA (and, to a lesser extent, inactivity) on BMI. Therefore, the last analysis evaluated the potential mediating role of PA and inactivity in relation to dietary behaviors and overweight adult children. Dietary behaviors were combined in whole scores based on positive and negative product consumption with the PCA method (eigenvalue of the first factor: 1.34 with a variance explained of 67.15%). Because the sex variable played a role in BMI status, the analyses were conducted separately for males (sons) and females (daughters). Logistic regression analysis was used at the beginning to test all covariates. The results are presented in Table 4.

The model for males was well fitted, as confirmed by the chi-square goodness of fit (χ^2^ = 31.23, *p* < 0.001) and contained significant dietary intake (*p* = 0.008) and PA (*p* < 0.001) behaviors, but there was no effect with sitting time (*p* = 0.086). Meanwhile, in females, only PA played a significant role in overweight status (*p* = 0.005) without the effect of dietary intake (*p* = 0.209) and sitting time (*p* = 0.948). This model was also well fitted (χ^2^ = 13.58, *p* = 0.004). Negative coefficients near IPAQ suggest increasing odds of being overweight together with less time spent being active. Similarly, the odds of being overweight increased with poor dietary behaviors.

The significant coexistence of dietary behaviors and PA in males confirmed the necessity of examining the potential mediating role of PA in the association between dietary behaviors and overweight status.

According to mediation procedures, two regression models were tested to investigate whether the association between dietary behaviors and being overweight is mediated by PA. In the first ordinary least squares regression model, dietary behaviors were non-significantly related to being overweight, b = 1.51, *p* = 0.83. In the second logistic regression model, which additionally included PA as a predictor of being overweight, neither dietary behaviors, b = −1.17, *p* = 0.008, nor PA, b = −0.04, *p* < 0.005, was significantly independently associated with overweight status. The bootstrap CI derived from 5000 samples indicated that the indirect effect coefficient was not significant, b = −0.05, CI = −0.520–0.319. These results showed a non-significant indirect effect because the bootstrap confidence included zero. Therefore, the result did not support the hypothesis that the relationship between dietary behaviors and overweight is mediated by PA in males. Both lifestyle factors are independent and significantly affect one’s overweight status.

In an analysis of females, dietary behaviors were not significantly related to being overweight, b = 4.10, *p* = 0.464. In the second logistic regression model, which included PA and dietary behaviors, b = −0.60, *p* = 0.209, was not significantly related to being overweight, while PA, b = −0.04, *p* = 0.005, was significantly associated with being overweight. The bootstrap CI derived from 5000 samples indicated that the indirect effect coefficient was not significant, b = −0.16, CI = −1.05–0.41. These results also did not support the hypothesis regarding the mediating role of PA in the relationship between dietary behaviors and overweight status in females. However, in this case, PA is a stronger lifestyle factor related to a female’s weight status.

## 4. Discussion

The main objective of our work was to explore parent–adult child dietary behaviors, focusing on patterns, potential intrinsic and extrinsic predictors of body mass, and determinants of being overweight and obese. The study revealed notable differences in dietary patterns between parents and children. Pairwise comparisons showed that sons exhibited a mix of positive and negative dietary behaviors, with fathers predominantly displaying negative behaviors; fathers of daughters exhibited mixed behaviors, while daughters mirrored their fathers’ positive behaviors, particularly in regard to the frequent consumption of fruits, vegetables, legumes, and fish. The dietary patterns of sons and mothers were more aligned, predominantly featuring positive behaviors such as higher consumption of vegetables, fruits, and a specific triad of fermented milk, milk, and wholegrain bread. In contrast, daughters and their mothers displayed mixed behaviors. In analyzing the predictors and statistical significance of pattern similarities, there was a divergence between parents and children, moderated by sex; there was a higher prevalence of similarities in overweight status among males and their parents, particularly fathers, with less prevalence of overweight concordance between daughters and their parents. Significant associations were found between non-healthy dietary intake (fast food, fried meals, alcoholic drinks, energy drinks, and sweetened beverages) and children being overweight. A predictor was also PA (inverse relationship), while not inactivity. PA and dietary behaviors independently determined children’s overweight status, particularly in males, with no observed mediating effect of PA.

This study showed similar frequencies of consuming healthy products and highlighted sex differences in the frequency of consuming unhealthy products. Males, unlike females, consumed fried meals, canned meals, sweetened carbonated beverages, and fast foods significantly more often, resulting in higher values on the non-healthy HDI index. These results align with several studies related to young adults, adolescents, and children [50,51], and partially agree with other Polish studies on young adults [52]. Women’s dietary behaviors more often correspond to healthy nutrition principles, with more meals consumed during the day, frequent consumption of fruits and vegetables, and the selection of lower-energy products.

The results showed a general divergence in dietary behaviors between adult children and their parents, with low similarities and weak strength for same-sex parent–child pairs. However, comparing sons to mothers and daughters to fathers, we found somewhat greater similarities and medium strength. Although children are influenced by their home food environment and community, they may have limited control over it [53]. Other studies found that the home food environment plays a stronger role in shaping a child’s intake of healthy foods than unhealthy foods, especially for young adults and adolescents [54]. The diversity between parental and adult children’s dietary patterns could be due to the frequency of eating out, which is influenced by changing home environments [13] and young adults starting their studies. The effect may be strengthened by freeing themselves from parental injunctions, prohibitions, or rewards for certain eating behaviors [4]. Although some studies have not found differences between mother and father food parenting practices and their impact on a child’s feeding [55,56,57], others found that fathers use more coercive food parenting practices compared to mothers [58,59,60].

Our study showed an affinity for family members concerning overweight patterns. Fathers’ body weight status was more similar to their daughters, and mothers to their sons, contributing to the overall overweight prevalence (26.6% and 29.7%, respectively). Our findings of overweight frequencies in adult children and their parents align with other studies. Arslan et al. (2021) [61] showed an obesity prevalence of 10.5% in children and 20.2% in their parents. Obesity was more prevalent in boys (12.0%) than in girls (9.0%), which coincides with our observations, although the proportion of overweight boys in our study was greater (12.66% vs. 36.92%). A lower prevalence of obesity in girls may be due to girls needing and consuming fewer calories than boys and being more conscious of their body shape [62,63]. Conversely, parental obesity was more frequent in females (20.4%) than in males, while our study showed more frequently overweight fathers (59.68%) than mothers (45.12%). The explanation could relate to the urbanization degree of the living area, although we did not analyze environmental factors.

Our results are in line with WHO data, where being overweight was higher in males; however, the obesity prevalence was higher in females [64,65]. High rates of overweight young adults agree with the Childhood Obesity Surveillance Initiative (COSI) by the Institute of Mother and Child in Warsaw, supported by the WHO Country Office for Poland, indicating over 30% of children and adolescents were overweight, and over 10% were obese [65]. More recent data from 2021 show increasing percentages of children and adolescents with excessive body weights, already exceeding 35%. Inappropriate body weight was more prevalent in boys (38.8%) than in girls (32.2%) [66].

This study revealed that fast food, fried meals, and unhealthy beverages (alcoholic, energy, and sweetened) were the most specific food products related to overweight status. These harmful effects are greater when consumed together. Studies on dietary behavioral changes during the adolescent–adulthood transition showed decreases in fruit and vegetable intake [23]. Among university students, those living in the parental home consumed healthier food compared to those living outside the parental home, including less fast food and unhealthy beverages [67]. Similar observations were made outside of Europe [68,69]. A relationship between non-healthy dietary habits, PA, and less inactivity was observed, aligning with other studies [70,71]. These covariates significantly affected BMI and overweight status, mirroring other results where non-healthy dietary behaviors were related to being overweight and obese, particularly in adolescents [72,73,74]. Reciprocal interactions between dietary behaviors and PA affect body fat, suggesting multidimensional effects of these lifestyle elements. Evidence from other studies suggests high PA coexists with better dietary behaviors, positively affecting body mass [75,76,77,78], supporting our findings.

A major strength of this study was its novel approach using cophylogenetic variants of clustering. Tanglegrams enhanced by cophenetic statistical analysis offers new insights into the interconnectedness of different behaviors. The juxtaposition of clustering trees allows simultaneous within-group and between-group analyses of associations between variables. To our knowledge, no other studies have used such an exploratory strategy in dietary behavioral associations.

The study has a few limitations. The cross-sectional design prohibits causal inference but enables the evaluation of associations. The small number of pairwise family affinities made in-depth examination difficult, despite the rule of thumb for cluster analysis confirming the validity of the calculations. However, normally, a sample size of >380 is used in cluster analyses. Recall bias (FFQ) may have affected the reliability and validity of survey findings. Participants were recruited from students at a university related to PA, making them unrepresentative of the broad young adult population. Thus, the results cannot be generalized further. The study included BMI data but no body weight composition data. Although we had body composition surveys of students, there was no such data on parents, so body fat mass or skeletal muscle mass was not included in the analyses. One should be cautious in inferring BMI without information on muscle mass. The high ethnic cohesion of the sample may also limit the applicability of the results across different cultures.

## 5. Conclusions

Adult children and parental dietary patterns are divergent, reflecting the influence of multiple factors on a child’s dietary habits during the transition to adulthood. However, this divergence is moderated by sex, as evidenced by closer similarities in dietary patterns. Reciprocal interactions between dietary intake—particularly positive dyads such as fruits and vegetables, fermented milk, and curd cheese—and PA significantly impacted the BMI of children. Parental negative dietary behaviors serve as significant extrinsic predictors of their children’s BMI.

Competing hypotheses include changes in eating habits due to increased independence as children start their studies, but a plausible explanation could also be a stronger pattern of negative eating behaviors. Parental support in forming healthy habits is crucial, as both PA patterns and dietary behaviors independently determine overweight status, especially in males. It is recommended that parents be incorporated into promotional programs to raise awareness of the need to simultaneously promote healthy dietary intake and PA.

Furthermore, studying dietary patterns in conjunction with PA, particularly in relation to the link between overweight children and overweight parents, presents a separate challenge. Research in this area should aim to more deeply understand these dependencies.

## Figures and Tables

**Figure 1 nutrients-16-02256-f001:**
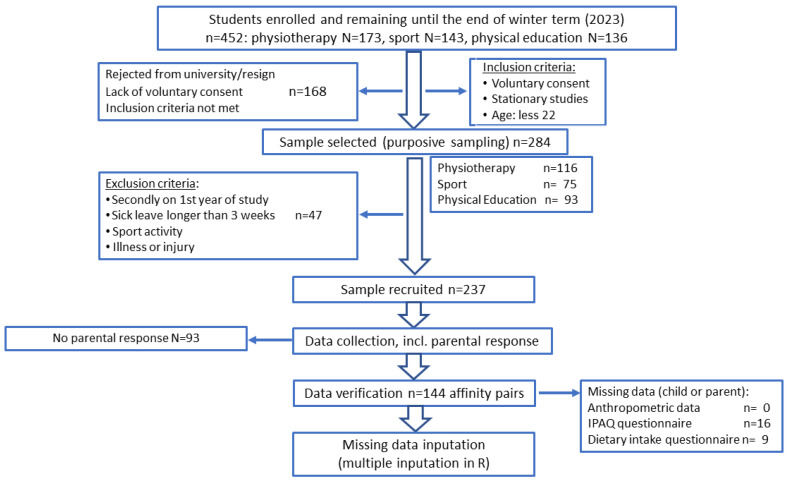
Flowchart: Study design and data collection.

**Figure 2 nutrients-16-02256-f002:**
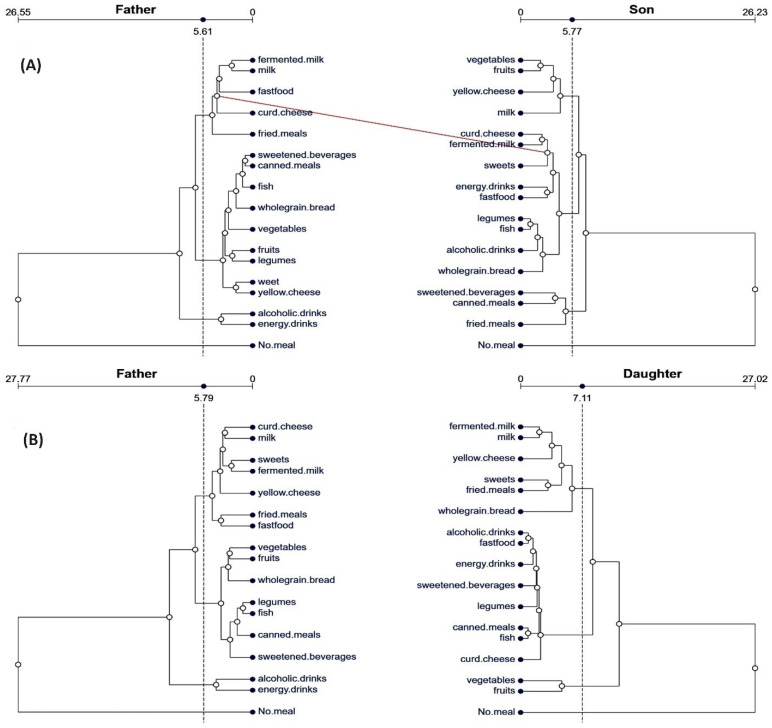
Tanglegrams of the dietary behavior patterns in family pairs: (**A**) fathers–sons, (**B**) fathers–daughters. Euclidean distances and Ward’s method of linkage were used. Primary dendrograms were untangled. The upper horizontal line presents the distances between items in each cluster. Dashed vertical lines show the distance thresholds. Red lines show the subset of different branches corresponding to the complement of the subset of equal branches.

**Figure 3 nutrients-16-02256-f003:**
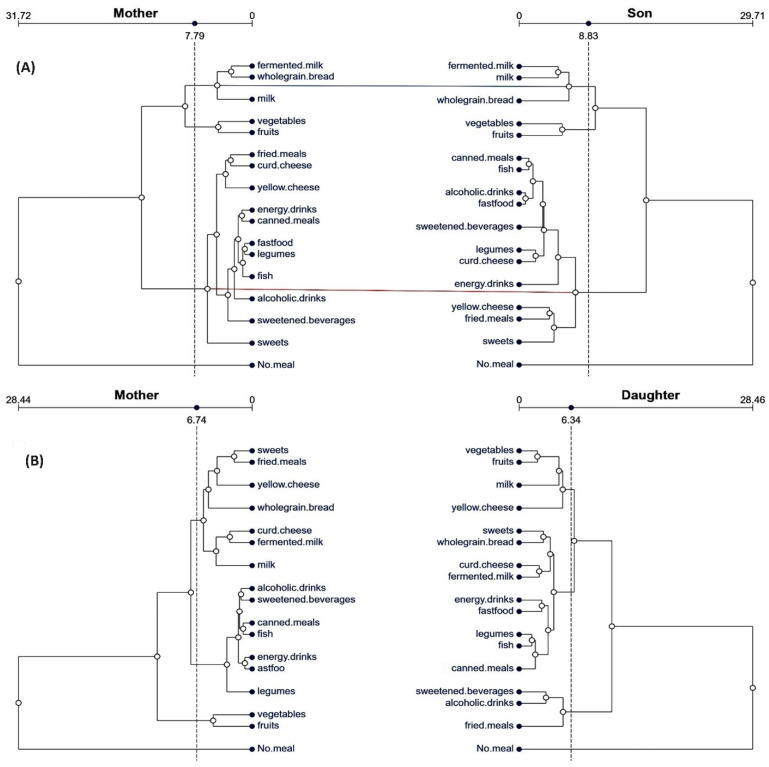
Tanglegrams of the dietary behavior patterns in family pairs: (**A**) mothers–sons and (**B**) mothers–daughters. Euclidean distances and Ward’s method of linkage were used. Primary dendrograms were untangled. The upper horizontal line presents the distances between items in each cluster. Dashed vertical lines show the distance thresholds. Red lines show the subset of different branches corresponding to the complement of the subset of equal branches.

**Figure 4 nutrients-16-02256-f004:**
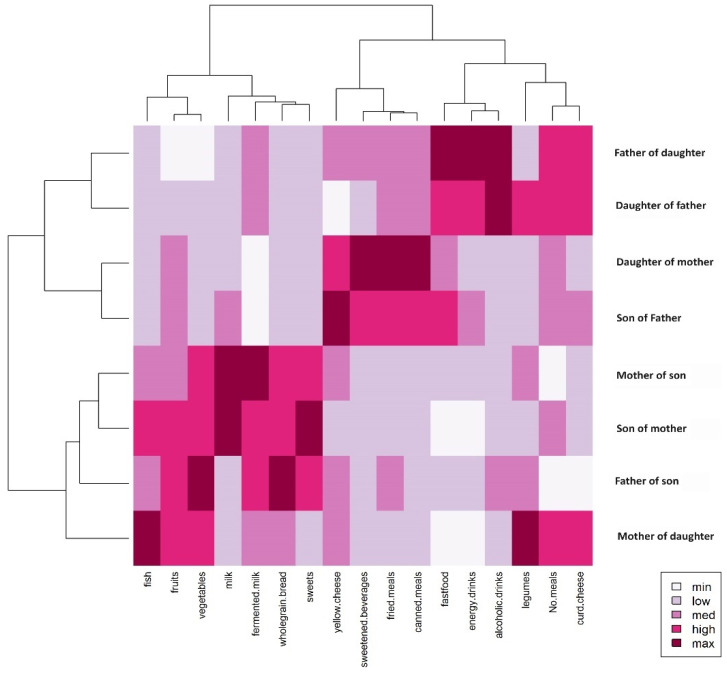
Two-way cluster analysis dendrograms (Ward’s method) with heatmaps (Mahalanobis distances) based on dietary behaviors (derived from the QEB questionnaire). The right side of the heatmap presents family groups, while the short phrases at the bottom represent dietary behaviors. The legend shows the strength of linkage—a darker color represents a shorter pairwise distance (between groups and dietary behaviors).

**Figure 5 nutrients-16-02256-f005:**
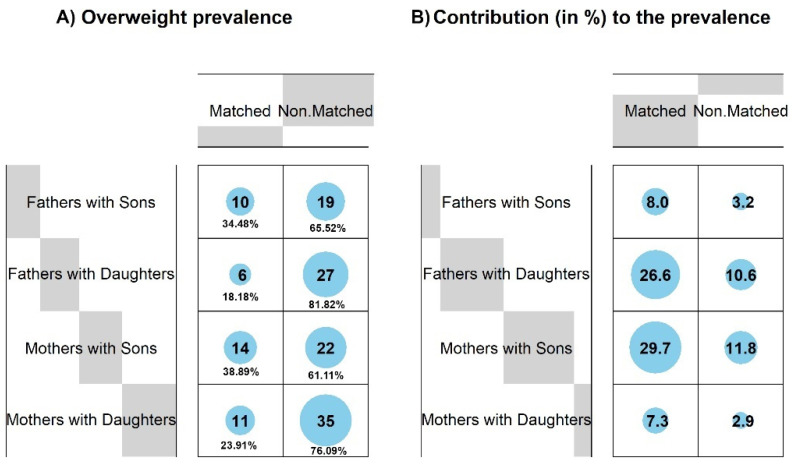
Balloon plots present a graphical display of the contingency table with numbers and frequencies of being overweight among parents and adult children (**A**) and the relative contribution of each cell to the total chi-square score (**B**). Results of the chi-square test: χ^2^ = 4.62, *p* = 202, C = 0.18.

**Figure 6 nutrients-16-02256-f006:**
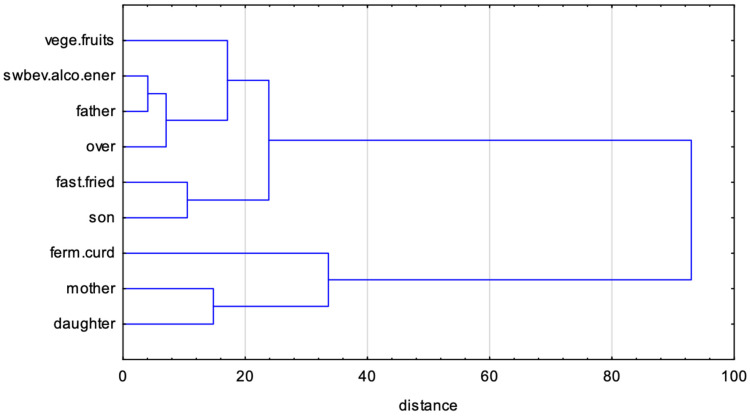
Dendrogram of the hierarchy of the 10 variables describing similarities between the family affinity group (father, mother, son, daughter), BMI status (overweight, normal weight), and dietary positive and negative behaviors combined in common indexes.

**Table 1 nutrients-16-02256-t001:** Characteristics (mean, SD, 95% CI) of the anthropometric, transformed dietary, and physical activity values of the adult children. F-values and *p*-values generated using ANOVA. Positive HDI, negative HDI, IPAQ, and sitting time results were transformed using the Yeo–Johnson power transformation.

			**Sons** **of Fathers**				**Daughters of Fathers**	
	**M**	**SD**	**−95% CI**	**+95% CI**	**M**	**SD**	**−95% CI**	**+95% CI**
Height [cm]	183.3	6.4	180.8	185.7	167.8	5.5	165.8	169.7
Weight [kg]	78.7	8.2	75.6	81.8	61.4	9.0	58.2	64.6
BMI [kg/m^2^]	23.4	2.6	22.4	24.3	21.8	3.1	20.7	22.9
Positive HDI	6.1	1.8	5.5	6.8	7.3	2.0	6.6	8.0
Negative HDI	6.9	2.1	6.1	7.6	4.5	1.0	4.1	4.8
IPAQ	159.7	48.7	141.2	178.2	143.0	48.6	125.7	160.2
Sitting time	86.3	17.2	79.8	92.9	88.0	15.0	82.7	93.3
			**Sons** **of Mothers**				**Daughters of Mothers**	
Height [cm]	182.8	6.7	180.5	185.1	167.9	5.3	166.3	169.5
Weight [kg]	80.4	10.3	77.0	83.9	61.4	9.7	58.6	64.3
BMI [kg/m^2^]	24.0	2.6	23.1	24.9	21.8	2.9	20.9	22.6
Positive HDI	6.0	1.7	5.4	6.6	8.8	1.3	8.4	9.2
Negative HDI	7.7	1.4	7.2	8.2	4.8	1.6	4.3	5.3
IPAQ	172.3	50.0	155.3	189.2	132.8	30.0	123.9	141.7
Sitting time	88.4	20.4	81.5	95.3	88.2	18.2	82.8	93.6
Significance F, *p*	**Height**: F = 76.40, *p* < 0.001; **Weight**: F = 44.96, *p* < 0.001; **BMI**: F = 5.81, *p* < 0.001; **Positive HDI**: F = 24.46, *p* < 0.001, **Negative HID**: F = 36.15, *p* < 0.001; **IPAQ;** F = 6.2, *p* < 0.001							

Footnotes: BMI—body mass index, positive HDI—positive (pro-) healthy dietary index, negative HDI—negative (non-) healthy dietary index, IPAQ—International Physical Activity Questionnaire, M—mean value, SD—standard deviation, CI—confidence interval.

**Table 2 nutrients-16-02256-t002:** Characteristics (mean, SD, 95% CI) of the anthropometric, transformed dietary, and physical activity values of the parents. F-values and *p*-values generated using ANOVA. Positive HDI, negative HDI, IPAQ, and sitting time results were transformed using the Yeo–Johnson power transformation.

			Fathers of Sons				Fathers of Daughters	
	**M**	**SD**	**−95% CI**	**+95% CI**	**M**	**SD**	**−95% CI**	**+95% CI**
Height [cm]	180.0	8.2	176.9	183.2	177.8	5.6	175.8	179.8
Weight [kg]	85.6	12.1	81.0	90.2	79.5	11.5	75.5	83.6
BMI [kg/m^2^]	26.4	3.4	25.1	27.7	25.1	2.9	24.1	26.1
Positive HDI	5.1	1.9	4.4	5.8	4.9	1.7	4.3	5.5
Negative HDI	6.9	2.0	6.1	7.7	7.5	2.2	6.7	8.3
IPAQ	160.4	46.6	142.7	178.2	162.6	53.1	143.8	181.4
Sitting time	89.1	15.2	83.4	94.9	92.2	17.2	86.1	98.3
			**Mothers** **of Sons**				**Mothers of Daughters**	
Height [cm]	169.0	5.0	167.3	170.7	167.5	5.5	165.8	169.1
Weight [kg]	70.7	11.7	66.7	74.7	70.2	12.0	66.6	73.8
BMI [kg/m^2^]	24.7	3.8	23.5	26.0	25.0	4.1	23.8	26.3
Positive HDI	8.3	1.7	7.7	8.9	8.4	2.0	7.8	9.0
Negative HDI	4.1	1.9	3.5	4.8	4.4	1.4	4.0	4.8
IPAQ	139.4	35.5	127.4	151.4	144.7	33.6	134.7	154.7
Sitting time	88.3	19.9	81.6	95.1	86.4	15.1	81.9	90.9
Significance F, *p*	**Height**: F = 37.50, *p* < 0.001; **Weight**: F = 13.29, *p* < 0.001 **Positive HDI**: F = 39.45, *p* < 0.001, **Negative HID**: F = 29.29, *p* < 0.001							

Footnotes: BMI—body mass index, positive HDI—positive (pro-) healthy dietary index, negative HDI—negative (non-) healthy dietary index, IPAQ—International Physical Activity Questionnaire, M—mean value, SD—standard deviation, CI—confidence interval.

**Table 3 nutrients-16-02256-t003:** Determinants of body mass index (BMI) in adult children in a stepwise backward linear regression.

Independent	b	95%CI	*p*-Value
Intrinsic (children’s)			
Sex	2.31	1.34–3.28	<0.01
ferm.curd	−0.55	−1.01–−0.07	0.024
IPAQ	−0.02	−0.03–−0.01	<0.01
Sitting (inactivity)	0.02	−0.00–0.05	0.082
**Accuracy**: R^2^ = 0.18, RMSE = 2.85, MAE = 2.25, AIC = 279.84			
Extrinsic (parental)			
vege.fruits	−0.29	−0.66–0.068	0.111
fast.fried	−0.44	−0.88–0.00	0.049
IPAQ	−0.01	−0.02–0.00	0.112
**Accuracy**: R^2^ = 0.10, RMSE = 2.90, MAE = 2.26, AIC = 311.97			

Footnotes: ferm.curd—fermented milk and curd cheese, IPAQ—International Physical Activity Questionnaire, vege.fruits—vegetables and fruits, fast.fried—fast food and fried meals, R^2^—determination coefficient, RMSE—root mean square error, MAE—mean absolute error, AIC—Akaike information criterion.

**Table 4 nutrients-16-02256-t004:** Determinants of obesity in males and females in a logistic regression analysis. The results are presented as regression coefficients (b), odds ratios (OR), 95% confidence intervals (CIs), and the significance of *p*-values.

Variable	b	OR	95%CI	*p*-Value
Males				
DI	−1.25	0.29	0.11–0.72	0.008
PA	−0.04	0.96	0.94–0.98	<0.001
Sitting time	0.03	1.03	1.00–1.08	0.076
Females				
DI	−0.61	0.55	0.21–1.40	0.209
PA	−0.04	0.96	0.94–0.99	0.005
Sitting time	0.01	1.00	0.95–1.05	0.948

Footnotes: DI—dietary behaviors (cumulative index of the pro-healthy and non-healthy dietary indexes), PA—physical activity (IPAQ overall scores).

## Data Availability

The data presented in this study is available on request from the author.

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
