# Peer review of "Sex-Moderated Divergence between Adult Child and Parental Dietary Behavior Patterns in Relation to Body Mass Condition—Evaluating the Mediating Role of Physical Activity: A Cross-Sectional Study"

_nutrients, 2024, doi:10.3390/nu16142256_

Round 1
Reviewer 1 Report
Comments and Suggestions for Authors
It is a very innovative article, although it is not well understood due to the lack of clarity in the writing and possibly because of the large number of objectives you have tried to address.
I have detected some grammatical errors (and I am not English) (ex. Line 64)
Abstract:
- Should be reduced
- adult children? That term seems ambiguous to me.
- Include the study design
- Results are very dense
Introduction: The introduction is very dense. The message you want to convey is lost. It is not clear whether you want to study mediators or obesity. There are too many objectives; reduce them to two at most.
Methods:
- Then, you haven't managed to meet the sample size?
- State clear that this is a cross-sectional study
- How did you choose whether a female participant (daughter) was included in “mothers of daughters” or in “daughters of mothers”? Check line 149.
- Methods usually starts with study design. Please, reorder the sections
- Participants: include exclusion and inclusion criteria
- What is HDI?
- Statistical analysis: extensive and detailed, but not explicative.
Results:
- A lot of tables and figures. Should be useful to include abbreviations in table foots.
- In my opinion, are difficult to understand. Should be clarify (similar as you do in the first paragraph of the discussion).
Discussion:
- Discuss the high rates of obesity prevalence that you found.
- Limitations should be completed: recall bias (FFQ), voluntary participation, participants from sports science (maybe better knowledge about diet and PA, thus, low generalizability of your results…) Please, reflect to them
Author Response
I would like to thank the reviewer for all the detailed suggestions, tips, and highlights that have significantly improved the overall quality of the article. In line with the Reviewer's comments, I have made numerous changes to the article. I send replies to each comment raised by the Reviewer. In addition, I have submitted an article with changes. I indicated any change made to the manuscript with tracking changes function.
I am sorry that the language of the article was not clear and caused difficulties in perception. In my excuses, I would like to add that it was sent for linguistic correction at a professional company that translates scientific texts. Unfortunately, I did not send the certificate. On the other hand, once I have made corrections to the Reviewer's comments, I will send the article for linguistic correction again and upload the certificate used.
It is a very innovative article, although it is not well understood due to the lack of clarity in the writing and possibly because of the large number of objectives you have tried to address.
I have detected some grammatical errors (and I am not English) (ex. Line 64)
1. Abstract:
-(1) Should be reduced
-(2) adult children? That term seems ambiguous to me.
-(3) Include the study design
-(4) Results are very dense
Answer:
- The Abstract was reduced.
- Adult children is commonly used term (in both colloquial and scientific language) to name persons who are adult, but compared to their parents. Unfortunately, I do not know of a term that would better replace the term adult children. In contrast, I have changed the term late adolescents to young adults, which for people in their 20s better conveys the meaning of age.
- I’d like to thank the reviewer for this comment. It’s very important information, that should not be omitted. ‘Non-probability, cross-sectional sampling was used to select participants from University stu-dents population.’ sentence was added.
- The results part of Abstract was reduces.
2. The introduction is very dense. The message you want to convey is lost. It is not clear whether you want to study mediators or obesity. There are too many objectives; reduce them to two at most.
Answer:
The Introduction was shortened. I agree with Reviewer, that too many objectives may overwhelm and paradoxically make the intentions less clear. Then, the bulleted objectives have been reduced to two. Although, the intention was to comprehensively cover several issues that are complementary and, in the authors' opinion, worth presenting in a single article, where the earlier analysis of dietary patterns is the context for the search for predictors of excessive body weight (in the analysis of direct and indirect effects).
3 . Methods:
- (1) Then, you haven't managed to meet the sample size?
-(2) State clear that this is a cross-sectional study
- (3) How did you choose whether a female participant (daughter) was included in “mothers of daughters” or in “daughters of mothers”? Check line 149.
- (4) Methods usually starts with study design. Please, reorder the sections
-(5) Participants: include exclusion and inclusion criteria
-(6) What is HDI?
- (7) Statistical analysis: extensive and detailed, but not explicative.
Answer:
- The most of cluster trees contain more or less four clusters, what which fits with the recommended number of 30 people per cluster. However, I agree, given that the clusters are made up of sub-clads, that larger numbers would give more credibility to the links. I have previously included such information in the Limitations section.
- It was added both in Abstract and Study design part of the work: ‘Non-probability, cross-sectional sampling was used to select participants from University stu-dents population.’
- It was clarified, that pairs of family affinity were studied: 46 pairs of mothers and daughters, 36 pairs of mothers and sons, 33 pairs of fathers and daughters and 29 pairs fathers and sons.
Participants included 288 healthy individuals, which were pairs of the family affinities: 46 were mothers and daughters, 36 were mothers and sons, 33 were fathers and daughters, and 29 were fathers and sons.
4. The subsections in the Method paragraph have been reordered.
5. Inclusion and exclusion criteria have been previously presented, however was supplemented with details.
Preliminary inclusion criteria were: voluntary consent, stationary studies and less than 22 years of age. The exclusion criteria were: secondly on 1st year of study, sick leave (longer than 3 weeks), sport activity and illness or injury.
6. QEB questionnaire questions were used to calculate two dietary indices: Prohealthy‐Diet‐Index (positive HDI) and Non‐Healthy‐Diet‐Index (negative HDI). It was supplemented in Dietary characteristics section.
7. The Statistics section is very extensive due to the multiplicity and complexity of the methods used. Especially when it comes to the non-standard study of patterns in the cophylogenetic approach. Such a description was intended to explain in detail the subsequent steps of the analytical procedure. We tried to balance between very detailed explanations, which would have stretched the description even further, and a description that would give an idea of the methods. Therefore, references to the literature, which provides details of the various procedures, were added.
4. Results:
-(1) A lot of tables and figures. Should be useful to include abbreviations in table foots.
-(2) In my opinion, are difficult to understand. Should be clarify (similar as you do in the first paragraph of the discussion).
Answer:
- I acknowledge that such an in-depth analysis as we have undertaken is difficult to briefly describe. We felt that it was better to provide more detailed and accurate descriptions rather than to write too briefly. We have provided the most necessary Tables and Figures necessary to document the analyses undertaken. We have dispensed with some earlier and only the basic statistics are presented in the text. Footnotes with the abbreviations were added to the tables.
- It is very difficult, without harming the documentation of the analyses, to shorten the Results. Only the most important information is included. Omitting them will result in a lack of logic in the subsequent steps of the analyses. That is why the first section in the Discussion contains a summary of the results leading to a comparison with the results of other authors and to conclusions. We ask the reviewer's forbearance for the lack of radical cuts to the results.
5. Discussion:
-(1) Discuss the high rates of obesity prevalence that you found.
-(2) Limitations should be completed: recall bias (FFQ), voluntary participation, participants from sports science (maybe better knowledge about diet and PA, thus, low generalizability of your results…) Please, reflect to them
Answer:
- Hight rates of obesity prevalence have been previously discussed in 3rd paragraph of the Discussion, however the results in Polish population was added. I’d like to thank the Reviewer for this suggestion.
- Thank you very much for this suggestion. Limitations were supplemented with suggestions.
Moreover, recall bias (FFQ) noted in present study may have directly affected the reliability and validity of survey findings. Furthermore, participants were recruited from students at the University related to physical activity. They are not representative of the broad population of young adults. Thus, the results and conclusions also cannot be generalized further.

Reviewer 2 Report
Comments and Suggestions for Authors
Dear author,
Thank you for sharing your research. I have a few suggestions to improve the manuscript:
1) The title should clearly state the research model employed.
2) It is unclear whether the study groups were randomized. Please clarify this point.
3) The formula for calculating the body mass index (BMI) is not necessary and should be removed from the manuscript.
4) The limitations section should include the potential influence of ethnicity on family relationships, as this may vary across different cultures.
5) The discussion is currently quite complex and dense, making it difficult for readers to follow the researchers' reasoning. It would be helpful to rephrase the discussion to make it more concise or to consider removing some data. A summary table of the results would also be helpful.
In addition to these specific points, I would also recommend that the manuscript be revised for overall clarity and conciseness. The use of jargon and overly technical language should be minimized, and the writing should be more engaging for a general audience.
Regards
Author Response
I would like to thank the reviewer for all the detailed suggestions, tips, and highlights that have significantly improved the overall quality of the article. In line with the Reviewer's comments, I have made numerous changes to the article. I send replies to each comment raised by the Reviewer. In addition, I have submitted an article with changes. I indicated any change made to the manuscript with tracking changes function.
Thank you for sharing your research. I have a few suggestions to improve the manuscript:
- The title should clearly state the research model employed.
Answer:
The subtitle indicating study design was added.
Sex Moderated Divergence Between Adult Child and Parental Dietary Behavior Patterns in Relation to the Body Mass Condition, Evaluating Mediating Role of the Physical Activity: Cro ss-Sectional Study
- It is unclear whether the study groups were randomized. Please clarify this point.
Answer:
It was added both in Abstract and Study design part of the work: ‘Non-probability, cross-sectional sampling was used to select participants from University students population.’
- The formula for calculating the body mass index (BMI) is not necessary and should be removed from the manuscript.
Answer:
The formula for BMI calculation was deleted.
- The limitations section should include the potential influence of ethnicity on family relationships, as this may vary across different cultures.
Answer:
Limitation section was supplemented with the text:
The next limitation is the very high ethnic cohesion of the sample, which varies across cultures and can be a hindrance in the discussion of the results obtained. The formula for BMI calculation was deleted.
- The discussion is currently quite complex and dense, making it difficult for readers to follow the researchers' reasoning. It would be helpful to rephrase the discussion to make it more concise or to consider removing some data. A summary table of the results would also be helpful.
Answer:
The discussion has been thoroughly revised. It has been shortened and rephrased. While, the summary table is not commonly used and, in order to thus avoid accusations of non-standard treatment of the article, we have chosen not to include such a table, but to summarize the results more clearly in the first section of the Discussion paragraph by pointing out the most important findings.
- In addition to these specific points, I would also recommend that the manuscript be revised for overall clarity and conciseness. The use of jargon and overly technical language should be minimized, and the writing should be more engaging for a general audience.
Answer:
The article have been sent to professional company to make needed language corrections.

Reviewer 3 Report
Comments and Suggestions for Authors
1. Line 56 I suggest updating "...but on the other hand, in long period of life, to..."...but also to..." or similar to improve sentence clarity.
2. Line 60 - consider rewording "fundamental area". I think this refers to "key predictor" or similar. The current wording is imprecise and will impact reader clarity.
3. Line 63 - update "are examined" to "have been examined".
4. Line 63-64 - I think that "The study are focused equally..." should read "Previous studies have focused both..." or similar for clarity and grammatical sense.
5. Line 75 - should "child obesity" be "childhood obesity"? Please check for clarity and flow.
6. Towards the end of the Introduction - it would be useful to provide greater context in the last 2-3 paragraphs to explain why this is important specifically in a Polish population. For example, are obesity rates high/increasing rapidly, or linked to higher degrees of risk than in other groups? Further, concise detail of public health challenges in Poland might help rationalise the stated Aims better for a wider readership.
7. Lines 107-122 - support statements made with appropriate citations in relation to most existing research focusing on younger children. Further (citation-supported) detail should be included to explain the sort of transitions that occur during starting university (presumably increased independence, students cooking/doing food provision for themselves, financial limitations, inexperience in dealing with all the above) that might again rationalise the approach here but also highlight the importance of these findings beyond the current target population.
8. Figure 1 - I think it should be "n = X" throughout this image, not "N = X". Also update to "Age <22y" or similar.
9. Methods/Discussion - have all parent height and weights also been collected in-person? Will this affect participation (e.g.. presumably only students with parents who live nearby will take part) and possibly findings (e.g. those with parents who live nearby are more likely to meet with their parents and eat together). Please check Methods are clearly described in terms of collection of parental data. There may be a need to discuss this approach further in your strengths and limitations section of your Discussion.
10. Methods - some further detail about the choice of food groups would be useful. For example, do these align with national food-based guidelines, or are these the groupings within the existing FFQ? Have all food groups available been tested, or was there a targeted approach undertaken?
11. Discussion - while the authors have rationalised the sample size from the potential to observe an association, there would appear to be major limitations in this current sample being representative (where normally a sample size of >380 is required) of the wider target population from which the sample has been selected. Please consider in your Discussion.
Comments on the Quality of English LanguageI have noted a number of sentences where clarity is an issue in my comments. Please check further.
Author Response
I would like to thank the reviewer for all the detailed suggestions, tips, and highlights that have significantly improved the overall quality of the article. In line with the Reviewer's comments, I have made numerous changes to the article. I send replies to each comment raised by the Reviewer. In addition, I have submitted an article with changes. I indicated any change made to the manuscript with tracking changes function.
- Line 56 I suggest updating "...but on the other hand, in long period of life, to..."...but also to..." or similar to improve sentence clarity.
Answer:
It was corrected as suggested.
- Line 60 - consider rewording "fundamental area". I think this refers to "key predictor" or similar. The current wording is imprecise and will impact reader clarity.
Answer:
It was corrected as suggested.
- Line 63 - update "are examined" to "have been examined".
Answer:
The sentence has been partially changed and deleted.
- Line 63-64 - I think that "The study are focused equally..." should read "Previous studies have focused both..." or similar for clarity and grammatical sense.
Answer:
The sentence has been changed.
- Line 75 - should "child obesity" be "childhood obesity"? Please check for clarity and flow.
Answer:
The sentence has been corrected.
- Towards the end of the Introduction - it would be useful to provide greater context in the last 2-3 paragraphs to explain why this is important specifically in a Polish population. For example, are obesity rates high/increasing rapidly, or linked to higher degrees of risk than in other groups? Further, concise detail of public health challenges in Poland might help rationalise the stated Aims better for a wider readership.
Answer:
One paragraph related to the Polish population and the increasing frequencies of the excessive weight was added.
- Lines 107-122 - support statements made with appropriate citations in relation to most existing research focusing on younger children. Further (citation-supported) detail should be included to explain the sort of transitions that occur during starting university (presumably increased independence, students cooking/doing food provision for themselves, financial limitations, inexperience in dealing with all the above) that might again rationalise the approach here but also highlight the importance of these findings beyond the current target population.
Answer:
This part was rephrased and deleted, according to the other reviewer’s suggestion.
- Figure 1 - I think it should be "n = X" throughout this image, not "N = X". Also update to "Age <22y" or similar.
Answer:
The Figure 1 was corrected.
- Methods/Discussion - have all parent height and weights also been collected in-person? Will this affect participation (e.g.. presumably only students with parents who live nearby will take part) and possibly findings (e.g. those with parents who live nearby are more likely to meet with their parents and eat together). Please check Methods are clearly described in terms of collection of parental data. There may be a need to discuss this approach further in your strengths and limitations section of your Discussion.
Answer:
This way of collecting information had no impact on bias. It can be assumed that living or not living with parents did not influence the return of the questionnaires. However, some other factors that we do not know decided that a much smaller group of parents responded positively than the number of students surveyed. A restriction referring to BIAS and FFQs was introduced in the Limiatations paragraph.
- Methods - some further detail about the choice of food groups would be useful. For example, do these align with national food-based guidelines, or are these the groupings within the existing FFQ? Have all food groups available been tested, or was there a targeted approach undertaken?
Answer:
A short information was added to the Dietary characteristics.
- Discussion - while the authors have rationalised the sample size from the potential to observe an association, there would appear to be major limitations in this current sample being representative (where normally a sample size of >380 is required) of the wider target population from which the sample has been selected. Please consider in your Discussion.
Answer:
Whole Discussion has been rewritten. Adequate limitation in relation to the number of the participants has been added.

Round 2
Reviewer 1 Report
Comments and Suggestions for Authors
Some minor comments:
Discussion:
First paragraph: overweight and obese
Bullet points should not be included in a discussion section
Thank you
Author Response
I'd like to thank the Reviewer once again for all comments which improved final version of the work.
Both suggestions were included. "Obese" was added and bullet points have been deleted.
Thank you.
Reviewer 2 Report
Comments and Suggestions for Authors
Dear Author, thank you. No changes needed.
Regards
Author Response
Thank you once again.
Reviewer 3 Report
Comments and Suggestions for Authors
Thank you for considering all points raised.
Comments on the Quality of English LanguageThere may still be some outstanding changes to make at proofing.
Author Response
Thank you very much once again for all comments and sugestion which improved significantly the final version of the work.